# A Machine Learning Approach for the Discovery of Ligand-Specific Functional Mechanisms of GPCRs

**DOI:** 10.3390/molecules24112097

**Published:** 2019-06-02

**Authors:** Ambrose Plante, Derek M. Shore, Giulia Morra, George Khelashvili, Harel Weinstein

**Affiliations:** 1Department of Physiology and Biophysics, Weill Cornell Medical College, New York, NY 10065, USA; agp2004@med.cornell.edu (A.P.); des2037@med.cornell.edu (D.M.S.); gim2010@med.cornell.edu (G.M.); gek2009@med.cornell.edu (G.K.); 2ICRM, Consiglio Nazionale delle Ricerche, 20131 Milano, Italy; 3Department of Physiology and Biophysics & Institute for Computational Biomedicine, Weill Cornell Medical College, NY 10065, USA

**Keywords:** functional selectivity, biased ligands, molecular dynamics, deep neural networks, sensitivity analysis, pharmacological efficacy

## Abstract

G protein-coupled receptors (GPCRs) play a key role in many cellular signaling mechanisms, and must select among multiple coupling possibilities in a ligand-specific manner in order to carry out a myriad of functions in diverse cellular contexts. Much has been learned about the molecular mechanisms of ligand-GPCR complexes from Molecular Dynamics (MD) simulations. However, to explore ligand-specific differences in the response of a GPCR to diverse ligands, as is required to understand ligand bias and functional selectivity, necessitates creating very large amounts of data from the needed large-scale simulations. This becomes a Big Data problem for the high dimensionality analysis of the accumulated trajectories. Here we describe a new machine learning (ML) approach to the problem that is based on transforming the analysis of GPCR function-related, ligand-specific differences encoded in the MD simulation trajectories into a representation recognizable by state-of-the-art deep learning object recognition technology. We illustrate this method by applying it to recognize the pharmacological classification of ligands bound to the 5-HT_2A_ and D2 subtypes of class-A GPCRs from the serotonin and dopamine families. The ML-based approach is shown to perform the classification task with high accuracy, and we identify the molecular determinants of the classifications in the context of GPCR structure and function. This study builds a framework for the efficient computational analysis of MD Big Data collected for the purpose of understanding ligand-specific GPCR activity.

## 1. Introduction

As our perception of G Protein-Coupled Receptors (GPCRs) evolve from simple on/off switches to multistate microprocessors [1], new questions arise regarding the complexity and specificity of GPCR signaling. A major reason for seeking this deeper understanding is the quest for information in support of rational design of more efficacious and specific drugs that exert their effects by targeting GPCRs. 

GPCRs are often the main drivers and regulators of signaling into the cell and must be able, therefore, to select among multiple signaling pathways based on the stimulus they receive for action. The ability of different ligands to elicit such differential signaling is commonly referred to as functional selectivity of the receptor, or biased agonism. For specific subtypes of serotonin (5-hydroxytriptamine; 5-HT) receptors, for example, Berg and colleagues [2] demonstrated that such biased signaling occurs through ligand-dependent coupling to G_q_ or G_i_ proteins. For other class-A GPCRs, biased agonism involves a selection between G protein and arrestin-coupled pathways. The notion that functional selectivity results from differences in receptor conformation arising from differential ligand-receptor interactions was proposed as soon as the dynamics of ligand-receptor complexes demonstrated the complexity of receptor activation [3,4,5]. Furthermore, structural evidence for distinct receptor conformations of angiotensin receptors activated by different ligands exhibiting functional selectivity has been reported [6]. But how a given ligand influences a receptor’s conformational ensemble remained enigmatic in spite of observations, e.g., for the 5-HT_2A_ receptor [7], of differences in some elements of the structural dynamics of receptors exhibiting functional selectivity. 

Molecular Dynamics (MD) simulation has proven to be a powerful tool for understanding protein function at the molecular level [8,9,10,11], and could deliver much-needed insight for the development of a molecular-level theory of functional selectivity of GPCRs. Notably, however, the study of functional selectivity with MD simulations would require the computation of many trajectories of GPCR-ligand complexes from extensive sets of ligands, each representing a functionally-selective class, in order to understand this allosteric mechanism [12]. Indeed, previous studies by Wingler et al. [6] suggest that the differences among the various ligand-dependent receptor conformations may be subtle. Thus, analyzing the MD data to gain insights into functional selectivity will require finding a low-density signal (i.e., detectable only after analyzing a lot of data) in a huge dataset, a common problem in the analysis of Big Data. Although there is no formally accepted definition of Big Data, it is generally referred to as “datasets that could not be perceived, acquired, managed, and processed by traditional (Information Technology) and software/hardware tools within a tolerable time” [13]. Concordantly, analyzing MD simulations of a very large scale remains a challenge and an open area of research. As discussed by Frankel and Reid [14], finding new ways to represent and interpret Big Data may allow us to explore new questions and extract new meaning from our research. These new representations must be efficient enough to process the data in a “tolerable” amount of time, and must be able to encode the relationship between the conformational details and functional properties of the molecules being studied.

The new frontier of analyzing Big Data lies in the field of machine learning (ML). Companies such as Facebook and Google, as well as academia (e.g., Massachusetts Institute of Technology), have already made great progress in this realm. For example, their algorithms excel at differentiating between highly similar objects, or identifying the same object in distinct states. Specifically, deep neural networks (DNNs) have already been shown to classify pixel representations of objects (pictures) with near-perfect accuracy [15]. The strengths of DNNs lie in their ability to find complex, non-linear patterns within data sets that may be too large and high dimensional for a human to analyze, or for which a model does not yet exist. It is reasonable, therefore, to consider the application of this class of approaches to the Big Data problem presented by the application of MD simulations to the analysis of GPCR mechanisms.

The difficulties in identifying conformational ensemble differences in GPCRs that are attributable to the binding of various ligands, fall into both categories mentioned above: the MD trajectories are too large and high dimensional to characterize manually, and there does not exist a sufficient structure/dynamics-based model of functional selectivity in which to understand them. Therefore, in order to use MD to advance such an understanding, we set out to transform the problem of identifying functional differences encoded in different structural states visited by MD trajectories, into a representation recognizable by state-of-the-art object-recognition technology. Since the effect of ligand binding to a functional site is the result of allosteric mechanisms [12,16,17] that alter the energetic landscape of the targeted protein, this new representation must differentiate between ensembles of ligand-dependent protein conformations. In GPCRs, information about the binding of ligands in either the orthosteric or allosteric sites is transferred through the receptor along allosteric pathways to a functional site on the protein [17]. It is reasonable, therefore, that in comparing ensembles of ligand-dependent conformations, these differences may involve only a small subset of residues. Consequently, in order to understand the molecular mechanism of protein function, we must be able to identify the motifs of the protein (residues/atoms) that undergo conformational rearrangements pertaining to ligand-specific functional states. The collection of these important motifs are known as Collective Variables (CVs). The identification of these CVs within the low-density signal of large-scale data is a challenging problem, but one which object-recognition technology is designed to tackle. 

To help achieve this goal of identifying the determinant motifs for the classification of ligand-specific functional responses of the receptor using an ML approach, we developed the algorithm described here that transforms MD trajectories into a representation readable by state-of-the-art object-recognition technology. This representation is fed into a pipeline that uses a Deep Neural Network (DNN) to: (i) recognize ensembles of molecular conformations and (ii) predict ligand-determined class according to structural differences learned from training. This pipeline then probes the DNN to identify CVs that are discriminative between the functional states of the receptor determined by the molecule being studied. We show how the results of this probing are then interpreted in the context of the molecular structure of the GPCR and its dynamic properties. To illustrate the application of this analysis pipeline, we present here its application to MD simulations of ligand-GPCR complexes in two receptor families, the serotonin receptor subtype 2A (5-HT_2A_R) and the dopamine receptor subtype D2 (D2R) bound to full, partial, and inverse agonists, a well-characterized pharmacological classification of receptor responses to ligands, that is expected to be recognized by the DNN. 

## 2. Results

### 2.1. Transformation of MD Trajectories into Pixel Representations Readable by DNNs

To apply image-recognition (picture-recognition) technology in the analysis of the MD trajectories for the various ligand-GPCR complexes, the complete MD trajectory for each complex must be transformed into a pixel representation that constitutes the canonical input for these ML algorithms. We developed a transformation that takes advantage of the similarities between the components of proteins and picture representations: both are composed of bits of information, i.e., the pixels in a picture and the atom in a protein. The definition of pixels in terms of their values of red, green, and blue (RGB) parallels the definition of the individual atoms by their coordinates X, Y, and Z (XYZ). Consequently, a unique representation of the protein as a picture is obtained when each atom of the protein is transformed into a pixel with an RGB value that corresponds to that atom’s XYZ value (Figure 1). 

Since this procedure is sensitive to the translational and rotational movements of the protein, the trajectories are pre-processed by scrambling to remove bias from the original trajectory (see the next section for details on the scrambling procedure).

### 2.2. Training and Application of DNNs Able to Recognize Ligand-Dependent Receptor Conformations

#### 2.2.1. General Protocol

The representation of a protein according to the rules described above in Section 2.1, encodes the three-dimensional structure of a molecule into a two-dimensional picture that can be fed directly into an ML algorithm, such as the convolutional neural network utilized here, without any loss of information (clearly, such a loss would occur if a conventional projection-like “picture” of the molecule were to be used). In the method of analysis presented here, this transformation is applied to each frame of the MD trajectory sequentially, and followed by the steps depicted in Figure 2. 

The trajectories collected from MD simulations of ligand-GPCR complex of the serotonin 5-HT_2A_R and dopamine D2R (see Methods for details) were used to illustrate the application of the pipeline described in Figure 2. The data submitted for the DNN contains the coordinates of the atoms in the receptor structures sampled at the time-steps (frames) of the MD trajectories, presented in the form of the visual representation of these coordinates (Figure 1), as well as the corresponding class label for each structure (i.e., bound to agonist, partial agonist, or inverse agonist).

The scrambling protocol applied to the data prior to submission to the neural network (NN) is an unbiasing step in which the position of each frame and its orientation are scrambled (randomly; see Figure 3 for more information on the trajectory scrambling). This is undertaken in order to eliminate from consideration by the NN any differences among frames that originate not from the time-dependent molecular dynamics, but from changes in position or orientation of the ligand-GPCR complex. Thus, the scrambling directs the NN algorithm to consider only the intramolecular changes of the protein induced by the ligands. This scrambling is introduced in our protocol to achieve the same unbiasing that is attained in image classification tasks by random orientation of objects in pictures (which forces the object recognition neural networks to understand the shapes and colors of objects, independent of their background and orientation). 

The convolution neural network (Figure 2) was trained on training and validation sets, and tested on the data set, following known ML protocols [15,18] (see below, Section 2.2.2). The results for the 5-HT_2A_R (Section 2.2.2) and D2R (Section 2.2.3) systems were then analyzed as described below in Section 3 to identify the molecular determinants of the classification.

#### 2.2.2. Pharmacological Classification of the MD Trajectories of 5-HT_2A_R-ligand Complexes

The set of 5-HT_2A_R-bound pharmacologically distinct ligands chosen for this illustrative application includes the full agonist serotonin (5-HT), the inverse agonist ketanserin (KET), and three partial agonists of differing efficacies (LSD, DOI, LIS), of which two are similar in molecular size (l-lysergic acid diethyl amide, LSD, and lisuride, LIS), and one is smaller, similar in size to 5-HT (2,5-Dimethoxy-4-iodoamphetamine, DOI). The data set consists of 43,565 structures sampled with a 0.2 ns time-interval from the full and inverse agonist trajectories, and a 0.6 ns time-interval from the partial agonist trajectories (to maintain a balance in the number of samples from each class). The complete set of trajectory data, along with their corresponding class labels, was split into training, validation, and test sets of size 24,396, 10,456, and 8,713, respectively. 

The 5-HT_2A_R complexes were subjected to the general protocol described in Section 2.2.1 above. The results for the 5-HT_2A_R test set quantify the accuracy of the neural network in predicting the class labels of pharmacological classifications of the 5-HT_2A_R ligands in the form of a confusion matrix (Figure 4). The formulation of the confusion matrix relates the predicted class label of the test set instances to their true class label. Each element of the confusion matrix is calculated as (Equation (1)):(1)$$C_{\left(i,j\right)}=\overset{N}{\underset{n=1}{\sum }}F_{n,\left(i,j\right)};\;where\;F_{n,\left(i,j\right)}=\{\begin{array}{l} 1,\;T_{n}=i\;AND\;P_{n}=j\\ 0,\;otherwise \end{array}$$
where $C_{\left(i,j\right)}$ is the (row, column) index of the confusion matrix, N is the number of instances in the test set, $T_{n}$ is the true class label of instance n, and $P_{n}$ is the class label of instance n predicted by the neural network. Each element is then normalized to the total number of instances in each class.

Results from the application of the analysis to the test set for the 5-HT_2A_R are presented in the confusion matrix in both numerical and a color-coded visual form Figure 4A. The diagonal elements represent the fraction of instances for which the neural network had correctly identified the class label. Off-diagonal elements show the fraction of incorrectly labeled instances and identify the class to which they were incorrectly attributed.

In this illustration of the method for the 5-HT_2A_R, >99% of the frames in the test set were correctly labeled (the diagonal of the confusion matrix in Figure 4A). This near-perfect accuracy achieved on the test set suggests that the neural network is able to recognize and classify the ligand-dependent receptor conformations presented by this data set, with most of the very few incorrectly labeled instances (the off-diagonal elements) involving confusion of partial agonist labels. The molecular determinants for this recognition are discussed in Section 2.3.

#### 2.2.3. Pharmacological Classification of D2R-Ligand Trajectories

In order to test the generalizability of the method to MD trajectories of other GPCRs, we analyzed trajectories of another class-A GPCR, the D2R, bound to pharmacologically distinct ligands, namely the full agonist (Dopamine, DA), inverse agonist (Sulpiride, SLP), and partial agonist (Aripiprazole, ARI). 

Following the same steps as described above for the analysis of 5-HT_2A_R, the D2R trajectories (see Methods) were subjected to the protocol described in Section 2.2.1 above. The visual representations of the coordinates from structures (frames) sampled every 0.4 ns from the trajectories of the ligand-D2R complexes were randomly split into training, validation, and test sets of size 20,017, 8580, and 7150, respectively. The accuracy of the neural network in predicting the class labels of the D2R ligands in the test set is presented in Figure 4B as a confusion matrix (see Equation (1) and Section 2.2.2 for details on how the confusion matrix was constructed). The same high-accuracy achieved by the neural network for the classification of the 5-HT_2A_R trajectories was attained in the classification of the smaller set of classes for the D2R system.

### 2.3. Identifying the Molecular Determinants of the Classification

#### 2.3.1. General Protocol

To reveal the identity of the molecular features that were most instrumental in the classification decisions of the NN, we employed a sensitivity analysis approach in the category of visual saliency [19]. This sensitivity analysis is based on computing the gradient of the neural network’s classification score for a particular label (i.e., how likely the network believes a picture to be in each class), with respect to each of the pixels of the input image. The higher the gradient for a particular pixel, the more attention the neural network paid to it in making the classification.

Figure 5 shows an example of this type of sensitivity analysis applied to a traditional image classification. Here a deep neural network has correctly labeled a picture as “dog”, and the part of the picture that the DNN utilized to make the classification is highlighted in the heat map (also called “attention map”) next to it that color-codes the magnitude of the gradient for each pixel. Clearly, the attention map in Figure 5 highlights (in the collection of highest gradient values) the pixels that correspond to the dog, designating them as those having been the most important pixels for classifying the picture correctly.

#### 2.3.2. Application of the Sensitivity Analysis to the 5-HT_2A_R Classifications

We applied this sensitivity analysis approach to identify the features used by the NN to correctly classify the test frames as those corresponding to the 5-HT_2A_R bound to a full agonist. Figure 6A shows an attention map calculated from the ligand-bound 5-HT_2A_R test set. This attention map was obtained by averaging the attention maps of 1000 instances (trajectory frames) of the full agonist bound 5-HT_2A_R. In this panel, the protein is represented by a 71 × 71 pixel matrix containing the atoms of the receptor protein (20 atoms are missing from the C terminus) and constructed as described in Figure 1. The elements are colored according to the color scheme with the largest gradient value representing the highest importance of the atom in the classification task. The calculation of the average takes advantage of the property of the visual representation of the protein in which each pixel always represents the same atom. 

To be able to relate the contributions of the identified key atoms and groups in the context of structural regions of the protein, we show in Figure 6B the same data as in the attention map (panel 6A), but in a different representation in which all the pixels (i.e., atoms) are listed on the X axis sequentially, labeled with their generic numbering that identifies the TM segments of the GPCR protein. Applying to Figure 6B an importance cutoff value of 0.15 (chosen by inspection of the attention map), the residues considered to be most important in the classification of the ligand bound to the receptor are found to reside in the middle and the extracellular ends of TMs 2–5 (residues at positions 2.56, 2.65, 3.28, 4.58, 4.59, 4.62, 4.63, 4.67, and 5.35–5.37, labeled with the generic Ballesteros-Weinstein numbering [20]), at the intracellular ends of TMs 2 and 3 (positions 2.38 and 3.55), and in the extracellular loops ECL1 and ECL2, as well as intracellular loops ICL2 and ICL3 (the pieces of which were identified as important exchange between loop and helical extensions of TM5 and TM6 throughout the trajectories). There is a roughly even distribution between important residues that reside in the loops and the helices. These sites are indicated on the 5-HT_2A_R structure in Figure 6C. 

To verify further the role of these specific structural motifs in the classification produced by the neural network, we compared the effect of blurring the pixels with the highest gradient (most important, salient) to those with the lowest gradient (unimportant). In this procedure, sets of pixels corresponding to a number N of least-important atoms for all classes, were blurred in the test set by replacing them with Gaussian noise. This erased the information that these pixels confer to the neural network. The classification score computed for such an altered image represents the probability of an image belonging to a particular class, and the sum over all classes is unity. The same blurring procedure was then repeated for the same number of pixels with the highest gradient (importance) for a particular class. 

The results of this analysis for the full agonist-bound class (in Figure 6D), show that when the most important pixels are blurred (blue dots), the classification score quickly decreases to the chance rate (~0.33 for 3 classes) with 250 blurred pixels. This means that the network is no longer able to identify this instance as bound to full agonist. However, after blurring the same and even larger numbers of unimportant pixels (orange dots), the neural network is still able to identify the instance as belonging to the full agonist-bound class (with a classification score >0.9). 

#### 2.3.3. Sensitivity Analysis Applied to the Classification of the D2R Complexes

The results of applying the sensitivity analysis to the pharmacological classification of ligand-bound complexes of the other class-A GPCR, D2R, as shown in Figure 4B, are summarized in Figure 7. Using the same importance cutoff value of 0.15 as in the analysis of the 5-HT_2A_R, we find that the most important residues in the classification of the ligand-D2R complexes reside in the middle and the extracellular ends of TMs 1, 4, and 5 (positions 1.29, 1.30, 1.33, 1.34, 1.37, 1.38, 1.42, 1.46, 1.51, 4.53, 4.62, 5.35, 5.36, 5.39, 5.40, and 5.44), at the intracellular ends of TMs 5-6 (positions 5.64, 5.69, 5.70, 5.72–5.74, 6.29, and 6.33), and in the extracellular loop ECL2, as well as intracellular loops ICL2 and ICL3. These sites are indicated on the D2R structure in Figure 7C. There is some striking overlap between the most important residues found by the attention map analyses of the D2R and 5-HT2_A_R trajectories. Among the most important regions found by both analyses was the intracellular ends of TM5 and TM6, the extracellular end of TM4 and TM5, as well as ECL2, ICL2, and ICL3. Remarkably, among the most important residues of both analyses was residues 4.62 and 5.36 on the extracellular ends of TM4 and TM5, respectively. Multiple analogous residues of the ECL2 as well were identified in both analyses, including the highly conserved cysteine (C227 in 5-HT2_A_R and C182 in D2R) that makes a disulfide bond with C3.25 and was shown to be critically important to the function of many class-A GPCRs [21].

### 2.4. Dynamic Differences Discriminated by the DNN in the Regions Most Important for the Classification Decisions of Ligand-Bound GPCRs

To gain additional insight into the nature of dynamic differences discriminated by the DNN in the regions found to be most important for the classification decisions, we compared the conformational dynamics of specific residues identified by the atoms with high importance for the classification. 

#### 2.4.1. 5-HT_2A_R Complexes

Starting with the most important region, as identified in Figure 6B, i.e., the second extracellular loop (ECL2), we evaluated the dynamic range of the top hit of the sensitivity analysis for the 5-HT_2A_R bound to the full agonist, compared to the aligned trajectories of the 5-HT_2A_R complex with a full agonist and the inverse agonist. Figure 8A shows the sampling of the *epsilon* carbon (Cε) atom of F222 in ECL2. Although the starting structures of the GPCR in all 5-HT_2A_R-ligand-bound complexes were highly similar (see Methods for details on docking), the conformational space sampled by this region diverged over the course of the simulations. As Figure 8A shows, ECL2 of the 5-HT_2A_R bound to the inverse agonist (KET) samples two distinct states over the trajectories in the dataset (red and orange points in Figure 7A), with the red points indicating conformations in which F222 is almost exclusively pointing inward towards the binding pocket and TMs 6 and 7 (see Figure 8B bottom left). In the other trajectory (orange points) of the KET-bound 5-HT_2A_R, a section of the ECL2, including F222, prefers an alpha-helical conformation, resulting in F222 pointing mostly away from the binding pocket towards the extracellular side (Figure 8B top left), with some brief overlap with the sampling of the red trajectory. Notably, both of these conformations are distinct from those sampled by the 5-HT_2A_R bound to the full agonist 5-HT (blue and purple points in Figure 8A). In the 5-HT-bound trajectories, the section of the ECL2 containing F222 samples disordered/unfolded conformations in which F222 primarily occupies a region of space away from the binding pocket than the KET-bound (orange) trajectory, but shifted closer to TMs 1 and 2. Interestingly, in the trajectories of the 5-HT_2A_R bound to partial agonists, the Cε of F222 samples regions of space that overlap with both of the regions sampled in the trajectories of the 5-HT_2A_R bound to full and inverse agonist.

On the intracellular side of the receptor, we followed the conformational dynamics of residue R310 in ICL3, containing the *epsilon* hydrogen (Hε) identified as a highly salient atom. Figure 9A shows the sampling of the Hε in the aligned trajectories of the 5-HT_2A_R bound to full 5-HT (in blue) and KET (in red). The two representative frames from the trajectories, shown in Figure 9B, illustrate the conformation of the ICL3 sampled by each class. There is near perfect separation between the space sampled by the Hε atom of R310 over the full and inverse agonist bound trajectories, confirming that this residue, identified as important by the neural network, indeed confers a large amount of information about whether and how the ensembles of ligand-dependent conformations differ. 

The conformations of the partial agonist trajectories are discriminated by a different set of residues. The ICL3 of the partial agonist is not differentiated by the R310 Hε, because in the partial agonist-GPCR complex this atom samples both of the spaces shown in Figure 9A, with some preference for the trajectory of the full agonist. Instead, Figure 10 shows that the conformation of the ICL3 in the partial agonist trajectories is different from the other complexes in that it preserves throughout the helical structure of the residue segment 306 to 310 of TM6, whereas the same piece of ICL3 samples a disordered/unfolded conformation in both the full and inverse agonist-bound trajectories (shown in Figure 9). This is evidenced by the sampling of T307 in the ICL3, a residue identified as important in the attention map of the partial agonist, but not the full agonist.

#### 2.4.2. D2R Complexes

For the dopamine receptor complexes we again focused first on ECL2, which was identified as important for the classification of the ligand-D2R complexes. Figure 11A shows the time dependence in the trajectory of the values taken by the dihedral angle across the disulfide bond formed between C182 in the loop and C3.25 (in TM3) over the full-agonist (DA)-bound and inverse agonist (SLP)-bound trajectories of the D2R. For the majority of the SLP-bound trajectories, the ECL2 samples conformations that are never sampled in the DA-bound trajectories. This is evidenced in the figure by the difference in the dihedral angle of the disulfide bond between the two cysteines. The DA trajectories (blue) remain in an angle range around −100°, never sampling above −40°, whereas the SLP (red) trajectory samples mostly around 100°. A simple cutoff of 0°, for example, can distinguish a majority of the inverse agonist frames. These differences in angle values determine different 3D orientations of the ECL2 which thus becomes a salient feature for describing the differences in the ensemble of conformations sampled by the trajectories of D2R bound to the full and inverse agonist.

Like in the 5-HT_2A_R complexes, the residues in the extensions of TM5 and TM6 are important for the classification by the DNN. For the dynamic context, we compared residues K5.70 and T5.74 (both identified as important by the D2R sensitivity analysis) of the extension of TM5 in the aligned trajectories of the ligand-D2R complexes. Interestingly, inspection of the trajectories shows a difference in the preference of helical conformation of the intracellular end of TM5 dependent on the backbone hydrogen bond between these residues. The full agonist-bound trajectories preferred the helical conformation while the inverse agonist-bound trajectories preferred a disordered conformation. Figure 12A shows a measure of the helicity involving residues K5.70–T5.74 in the trajectories, quantified by the distance between the backbone oxygen of K5.70 and the backbone amide hydrogen of T5.74. The majority of frames in the DA-bound trajectories exhibit a smaller distance than those of the SLP-bound trajectories, because K5.70 and T5.74 form part of a helical structure for most of the full agonist trajectory, whereas in the inverse agonist-bound trajectories these residues are in a more disordered conformation. Figure 12B presents example frames from the ensemble of conformations sampled by D2R bound to the full and inverse agonist, underscoring the clear difference in helicity of the TM5 region and thus the structural context of the identified importance of the two residues in the classification by the DNN. 

## 3. Discussion

The machine learning-based approach we developed and illustrated here with applications to the classification of ligand-determined GPCR conformational properties points to some new directions taken by methods for MD trajectory analysis. In particular, analyses involving machine learning are growing in popularity as a greater variety of scientific questions are being addressed with MD simulations, and the systems targeted are increasingly complex. The data generated by these ever larger scales of simulation, accrued for large systems that require longer simulation times, is moving the field into the realm of Big Data Science [22]. A recent review of fundamental MD problems that could be addressed in the framework of Machine Learning (ML) [23], shows how ML has the potential to tackle problems arising from large-scale simulations, and that it has already had a profound impact on alleviating them (see Noe et al. [23] and references therein). 

The work we present supports the overall optimistic outlook regarding the potential of ML algorithms to extract important functional information from MD simulation trajectories in the usual structural and dynamic context of molecular mechanisms. An imperative in this respect is the development of approaches to translate the results obtained with DNN-based protocols into clear structural information in order to illuminate the underlying mechanisms in a physics-based context. By extracting automatically the functional information we achieved a major reduction in dimensionality of MD Big Data into a collection of salient structural motifs that efficiently describe the differences between ensembles of conformations sampled by each tested state. The results of this study are particularly encouraging, despite the limited scope of the test framework used in the illustration, as the approaches (i) classify correctly the ligand-bound GPCRs and (ii) identify the molecular determinants of the classification, and thus also the dynamic differences in the regions found to be most important for the classification decisions. 

Only 5 ligands for the serotonin 5-HT_2A_R and 3 for the dopamine D2 receptors were used in this limited test, but the results show that through the methodology described herein, a DNN is able to identify the relation between the functional properties of ligand-receptor complexes and the ensemble of conformations that they sample within MD trajectories. Due to the limited sampling of both configurational space and the classification field, the trajectory data served here only to exemplify the ability of the method to distinguish functional states of a GPCR, not to reach any conclusion about the significance of the functional states themselves. And yet, it is encouraging to observe that most of the salient regions for identifying the ligand-dependent conformational differences are common to both the 5-HT_2A_R and D2R, and are regions known to be important for GPCR activation. Thus, the ECL2 has been shown to play a critical role in class-A GPCR activation [24], taking on ligand-dependent conformations that are important for function [24,25,26,27]. The regions of the intracellular side of the receptor identified by our analysis, including the two ends of the ICL3 (i.e., the N-terminus of the loop which continues TM5, and its C-terminus which elongates the start of TM6), are also critically important for GPCR activation as these sites directly interact with the G protein [28]. As such, they are involved in the opening, observed in active GPCRs [29], of the intracellular side of the receptor. 

It is noteworthy, however, that while they serve to discriminate between ligand classes, the dynamics of these salient motifs differ between the 5-HT_2A_R and D2R systems. This is further reassuring because even if their role in recognizing the G protein, say (for the ICL3 motifs) is the same, they must respond to different G proteins in different functional contexts. However, that the structural motifs mentioned above were similarly sensitive to the bound ligand in both systems is concordant with the evidence that class-A GPCRs involve similar structural motifs in the pharmacological response. Collectively, the accuracy and agreement between the sensitivity analyses applied to the 5-HT_2A_R and D2R trajectories suggests that this method should generalize to other GPCR simulation studies, including trajectories of other ligand-GPCR complexes in diverse coupling pathways.

The characteristics of the novel method of analysis described here support its applicability not just to classifications of receptor activity (efficacy), but by finding ligand-dependent differences in MD trajectories of GPCRs, it is especially promising for the classification of ligand bias in functional selectivity as well. Our ongoing work is exploring this application, as well as other functional classifications being studied such as membrane lipid composition or the presence or absence of an action such as ion release.

## 4. Materials and Methods

### 4.1. Building Homology Model of the GPCRs

MODELLER (v.9.18) [30] was used to generate the sets of homology models of the human 5-HT_2A_R, and the human dopamine D2R. Briefly, Modeller generates 3D homology models of proteins using sequence alignments between the target and homologous proteins, as well as experimental structural data (e.g., x-ray or cryoEM structures from homologous proteins). These models are generated by optimally satisfying spatial restraints that are derived from the template structure(s). Because template-derived spatial constraints inform model construction, there is tension between choosing the most homologous template structures and providing Modeller with a set of template structures that possess enough sequential/structural heterogeneity so that the resultant set of homology models are significantly structurally diverse.

*Modeling the 5-HT_2A_R:* Three sets of sequence alignments/crystal structures were used as templates in Modeller: 

Set 1: consisted of two structures of the human 5HT_2B_R (PDBID: 4ib4 and 5tvn); 

Set 2: included two structures of the human 5HT_2B_R (PDBID: 4ib4 and 5tvn) and two structures of the human 5HT_1B_R (PDBID: 4iaq and 4iar);

Set 3: included all the structures in Set 2, augmented by 2 structures of the human ß2-adrenergic receptor b2AR (PDBID: 4lde and 4ldl).

Each of these template structures includes the receptor bound to one of its agonists. For each template set, Modeller was used to generate 1000 homology models of the 5-HT_2A_R. To select a single homology model from the 3 sets of Modeller outputs, each model was evaluated for its ability to discriminate between 5-HT_2A_R agonists and decoy ligands (i.e., ligands that are predicted to not bind 5-HT_2A_R). This characterization was performed by employing a multistep procedure. 

First, a set of 47 known human 5-HT_2A_R agonists was obtained from the IUPHAR/BPS database [accessed on May 1st, 2017] [31]. These agonists were then submitted to the DUD-E server [accessed on May 5th, 2017] [32], which generated a set of 3229 decoy structures. Briefly, DUD-E generates decoy structures that have similar physical chemistry properties (e.g., molecular weight, hydrogen bond donors/acceptors, rotatable bonds, etc.) but have a dissimilar topology. Schrodinger’s LigPrep software (v. 40015) was used to convert the agonist set and DUD-E SMILES descriptions into 3D ligand models. 

Next, the 3 sets of 1000 Modeller homology models were prepared for docking studies using Schrodinger’s Protein Preparation Wizard (v. 2016-2) [33]. This procedure added hydrogen atoms and created the disulfide bond between C3.25 and ECL2 loop residue C227. Schrodinger’s Glide software (v.34014) [34] was used to dock each of the 47 agonist and 3229 decoy structures into the models in the 3 sets (3000 models total), using standard precision. For each receptor/ligand complex, Glide calculates a ‘GlideScore’, which is an approximation of the ligand’s free energy of binding to the protein. 

Each model’s ability to discriminate between true agonist and decoy structures was quantified using the set of ligand GlideScores. The set that only used 5HT_2B_ as input templates was the most discriminating model overall and was used for subsequent docking studies, described below.

### 4.2. Molecular Construct for D2R

The crystal structure of the D2 dopamine receptor bound to Risperidone [27] was used as starting construct. The missing intracellular loop IL2 was modelled using MODELLER [30]. The long intracellular loop IL3, substituted by T4 lysozyme in the crystal construct, was replaced by a 10mer of Gly residues, inserted between S229 and Q365.

### 4.3. Parametrization and Docking of the Molecular Models

MOL2 files for the 5-HT2_A_R ligands (5-HT, Ketanserin, LSD, Lisuride, and DOI) as well as for the D2R ligands (Dopamine, Sulpiride, Aripiprazole) were obtained from the ZINC [35] database. The amide nitrogen in these compounds was protonated and docking poses were generated into the identical 5-HT2_A_R homology model and into the D2R structure using the Induced Fit [36,37,38] protocol in the Schrodinger Suite. A starting binding pose was chosen for each ligand based on the e-modal score and comparison to experimental data [25,26,27,39,40,41,42,43,44,45,46]. The starting binding pose for each ligand was chosen by comparison to experimental data as follows:

**5HT**—Experimental evidence suggests that serotonin interacts with S3.36 [40], F6.52 [39], D3.32 [40], S5.46 [44] and F6.51 [39]. The initial binding pose selected from the docking of 5HT satisfied these interactions.

**DOI**—Experimental evidence suggests that DOI interacts with F6.52 and D3.32 [40]. The initial binding pose selected from the docking of DOI satisfied these interactions, and also interacted with S3.36 (no literature studying this interaction, but it seems reasonable given the similarity of the amine group to that of 5HT which is known to engage with this residue). 

**LSD**—Experimental evidence suggests that LSD interacts with S5.46 [44], F6.52 and F6.51 [39], and D3.32 [40]. The initial binding pose selected from the docking of LSD satisfied these interactions and agreed with the shape and interactions seen in the crystal structure of LSD bound to the 5-HT_2B_ receptor [26].

**LIS**—Experimental evidence suggests that LIS interacts with S5.46 [44], F6.52 and F6.51 [39], and D3.32 [40]. The initial binding pose selected from the docking of LIS satisfied these interactions.

**KET**—The starting pose for ketanserin was selected by comparison to a study on the interaction of the inverse agonists risperidone and ketanserin with the 5HT_2A_R [45] as well as the crystal structures of risperidone, in complex with the D2R [27] and the 5-HT_2C_R [25]. These suggested that ketanserin interacts directly with D3.32 and W6.48. The starting pose for ketanserin satisfied these interactions and was similar to the pose of risperidone in the crystal structures. 

**DOP**—The starting pose has the maximum docking score among those conserving the interaction between D3.32 and the amide moiety of DOP, as well as hydrogen bonding to S5.42 and S5.46 [47]. In this arrangement, the aromatic ring of DOP comes close to W6.48.

**SLP**—In the binding pose chosen from the docking, sulpiride has an arrangement such that the protonated amine forms a salt bridge with Asp3.32, in agreement with ref [42], and similar to the crystal structure of eticlopride bound D3R [46], while the aromatic moiety occupies the hydrophobic pocket near the aromatic residues of TM6. 

**ARI**—For aripiprazole, the selected binding conformation has the dichlorophenyl ring oriented towards S5.42 and TM6 aromatic residues (W6.48), as proposed in previous studies [43], whereas the protonated amine interacts with D3.32 and the bicyclic heterocycle on the opposite side extends towards Y7.35.

Parameters for the CHARMM36 [48] force-field for each ligand were obtained by analogy from the CGenFF program [49,50]. Parameters with analogy penalties greater than or equal to 5 were optimized using the Force Field Toolkit (ffTK) [51]. Quantum mechanical calculations were made at the Hartree-Fock level of theory using the Gaussian software [52]. 

### 4.4. Molecular Dynamics Simulations

The GPCR systems were simulated in atomistically explicit membrane environments following established lab protocols (e.g., see REFs). For the simulation of 5-HT_2A_R-ligand complexes the molecular model was inserted into a membrane containing 144:16 POPC:Cholesterol molecules of in each leaflet. The membranes were built using the CHARMM-GUI [53] and equilibrated using the NAMD [54] software ver2.12 according to an equilibration protocol generated [55] by the CHARMM-GUI. After inserting the protein into the equilibrated membrane, the protein-membrane complex was surrounded by a 0.15 M NaCl solution with a hydration number of 80 water molecules per lipid using the CHARMM-GUI. Similarly, for the D2R-ligand simulations the structures were embedded, using the CHARMM-GUI, into a membrane consisting of 180:18 POPC:Cholesterol molecules per leaflet, and was solvated and ionized using the same conditions as above. 

Each complete system was equilibrated under the NPT ensemble (T = 310 K) in NAMD according to a previously established multistep equilibration protocol [16,56] before running approximately 1.5 microsecond long trajectories for each under the NVT ensemble (T = 310 K), except for the 5-HT_2A_R-DOI complex. After the equilibration protocol, five trajectories of the 5-HT_2A_R-DOI complex were run for ~400 ns before being split evenly into 25, ~100 ns long trajectories. The production runs were carried out with the ACEMD3 [57] software with a 4 femtosecond time-step and with all the standard simulation parameters [5,9,16,56] following established and documented protocols. 

The evolution of root mean square deviations calculated for the alpha carbon and ligand heavy atoms over the trajectories is shown in Appendix A. 

### 4.5. Machine Learning Procedures

*Building the DNN:* A custom Densely Connected Neural Network [15,18] was constructed, with 4 dense blocks containing 6, 12, 36, and 24 layers respectively, a growth rate of 48 filters per layer, 96 initial filters, and a reduction ratio of 0.5, in Keras [58] with a Tensorflow [59] backend based on an established implementation [60].

*Classification of the Trajectories:* Each frame of the scrambled trajectories was converted into a visual representation using the previously-described algorithm. Each frame was assigned a label according to its functional class (0 for full agonist bound, 1 for partial agonist bound, and 2 for inverse agonist bound). The frames were randomly split into a training, validation, and test set in the ratio of 56:24:20, respectively. The neural network was trained on the training and validation sets, and tested on the test set. The reported score is based on the test set. 

*Sensitivity analysis (keras-vis):* The sensitivity analysis was performed by computing the gradient using the visual saliency package provided by keras-vis [61] with guided back-propagation.

## Figures and Tables

**Figure 1 molecules-24-02097-f001:**
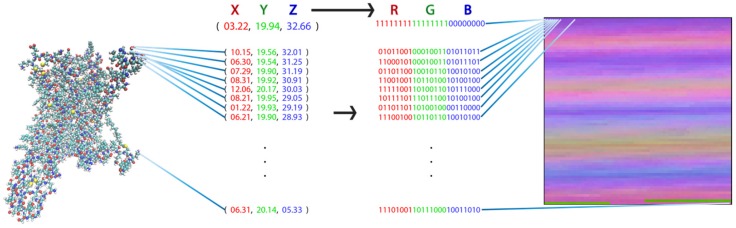
Visual representation of a molecular structure. Each atom of the molecule (left) is identified by the set of (X,Y,Z) coordinates as illustrated by the numerical set. The transformation to a 2D picture-like representation is obtained by assignment to each pixel representing an atom (in sequential order from top left to bottom right) by a pixel whose red, green, blue (RGB) value is the XYZ coordinate of the atom it represents (identified by the set of digital values). This representation has the special property that each pixel (i.e., matrix element) always represents the same atom in each frame from the trajectory of a particular protein.

**Figure 2 molecules-24-02097-f002:**
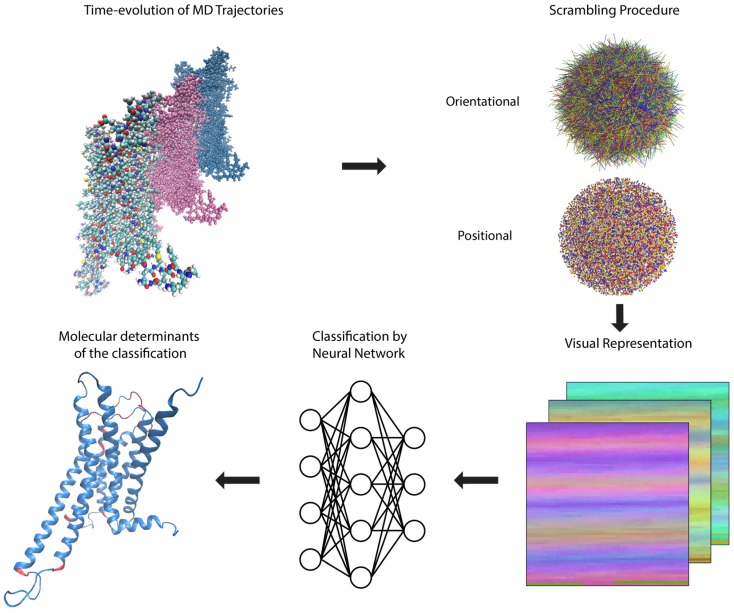
Overview of the computer-aided inspection pipeline described in detail in the text. Molecular Dynamics (MD) trajectories representing the time evolution of each system are subjected to a scrambling procedure to erase positional and orientational information and then each frame is converted into a visual representation suitable for feeding to a deep convolution neural network for training, validation, and testing of classification accuracy. A sensitivity analysis protocol is carried out to reveal the most important parts of the molecule (highlighted in red, see text and subsequent figures for more details) used in the classification by the neural network, and collective variables are extracted from the network to aid in further analysis.

**Figure 3 molecules-24-02097-f003:**
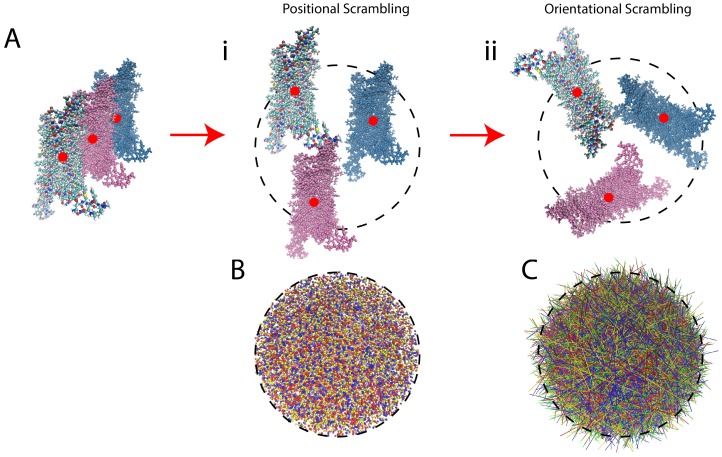
Illustration of the protocol for scrambling the frames of the MD trajectories. (**A**) Panel **i** Illustrates the positional scrambling of frames from a trajectory (depicted by the colored versions of the protein) by moving the centers of mass (red dots) to a randomly sampled coordinate within a sphere of diameter 90 Å (the size of the largest dimension of the receptor, indicated by the dashed circle). Panel **ii** Illustrates the orientational scrambling of the receptor by aligning a vector defined by two arbitrarily chosen atoms (preferably on an axis connecting the intracellular and extracellular ends of the GPCR) to a random unit vector in spherical coordinates. (Note: for the 5-HT2_A_R, the two atoms chosen were atoms 3695 and 4346, which are the eta hydrogen (Hη) of K6.32 and one of the delta hydrogens (H_δ_) of L7.34, respectively). (**B**) Illustrates the arbitrary positioning in space of the frames belonging to different trajectories of different receptor-ligand complexes, by showing the center of mass of the first 1000 frames of arbitrarily-chosen trajectories from each class. Frames from a full agonist-bound trajectory are in blue, from partial agonist-bound are yellow, from inverse agonist-bound are red. (**C**) Illustrates the arbitrary orientation of the frames from the different trajectories: green-tipped arrows with shafts going through atoms 3695 and 4346 are colored by the same color scheme as in **B**, and drawn for each of the first 1000 frames of the same trajectories as in panel **B**.

**Figure 4 molecules-24-02097-f004:**
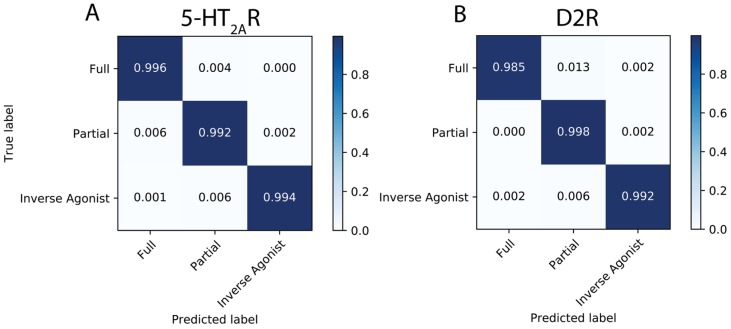
Results of the classification for the test set of the 5-HT_2A_R (**A**), and D2R (**B**). The coded coloring and numbers show the proportion of times the neural network predicted correctly a class label (diagonal elements), and the extent to which it failed to correctly predict the class label (off-diagonal elements) by confusing it with another class.

**Figure 5 molecules-24-02097-f005:**
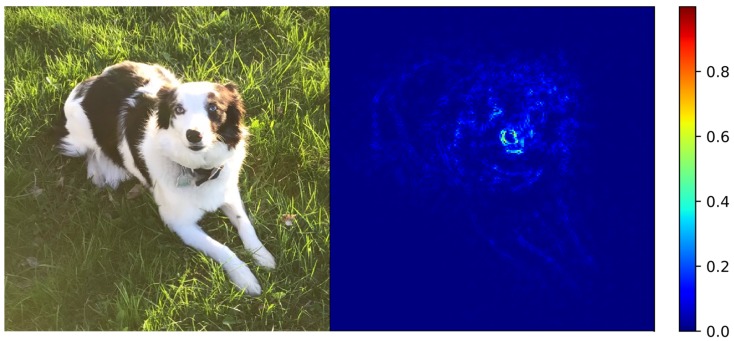
Sensitivity analysis applied to an image of a dog. For the heat map (attention map) on the right the color bar quantifies the gradient of each pixel with respect to how likely the neural network considers this image to be of a dog. This highlights the pixels that the neural network considered to be the most important for the classification of the picture. The pixels that correspond to the outline of the dog are seen to be the most important for the classification and thus have the highest gradient.

**Figure 6 molecules-24-02097-f006:**
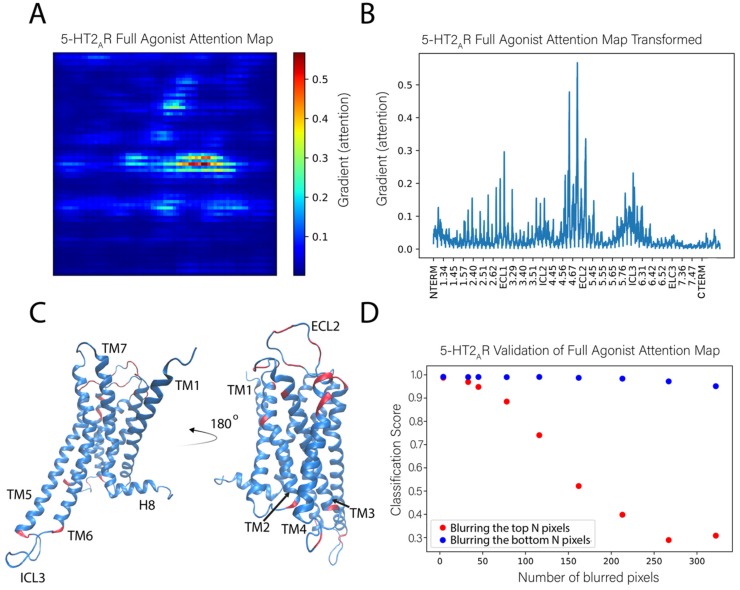
Identification of the molecular determinants of the classification of the full agonist bound to the 5-HT_2A_R. (**A**) The full agonist attention map shown was obtained as the average of 1000 attention maps calculated from 1000 instances of the full agonist bound to the 5-HT_2A_R. (**B**) Representation of the (average) attention map in panel A in which all the pixels (i.e., atoms) are listed on the X axis. The tick marks on the X axis identify residues labeled with the generic Ballesteros–Weinstein numbering [20]. The gradient values calculated for the classifier for individual atoms is on the Y axis, it indicates the order of importance (salience) of each atom for the classification. (**C**) The positions of the most important residues for the classification are indicated (in orange) on the ribbon representation (in blue) of the 5-HT_2A_R model structure. The highlighted residues contain atoms whose gradient (attention) is >0.15. (**D**) Validation of the attention map by evaluation of the effect of blurring increasing numbers of pixels (atoms) in the representation of the protein complex. Orange dots indicate the results of blurring increasing numbers of pixels identified to have the lowest gradient (least important). Blue dots indicate the results of blurring pixels with the highest gradient (most important). The number of atoms blurred is shown on the X axis while the Y axis shows the corresponding classification score of the full agonist for the blurred images.

**Figure 7 molecules-24-02097-f007:**
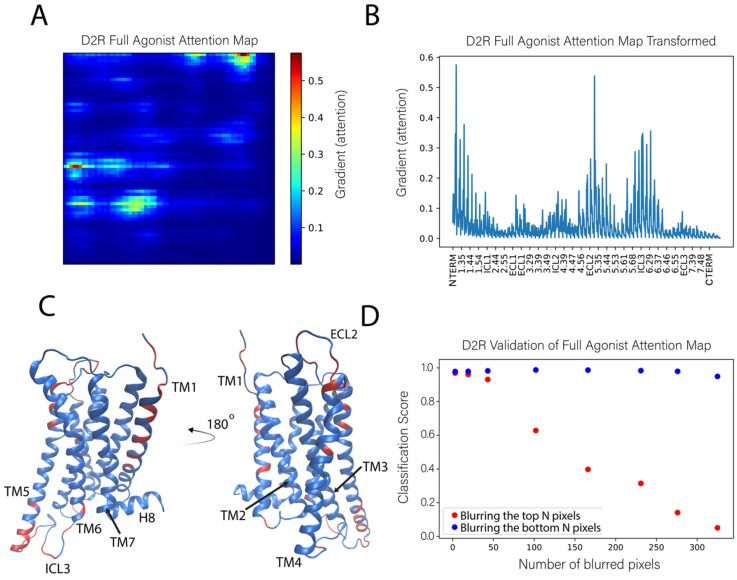
Identification of the molecular determinants of the classification of the full agonist bound to the D2R. (**A**) The full agonist attention map shown was obtained as the average of 1000 attention maps calculated from 1000 instances of the full agonist bound to the D2R. (**B**) Representation of the (average) attention map in panel A represented as in Figure 6B. (**C**) The positions of the most important residues for the classification are indicated (in orange) on the ribbon representation (in blue) of the D2R model structure. The highlighted residues contain atoms whose gradient (attention) is >0.15. (**D**) Validation of the attention map following the same procedure as described in Figure 6D.

**Figure 8 molecules-24-02097-f008:**
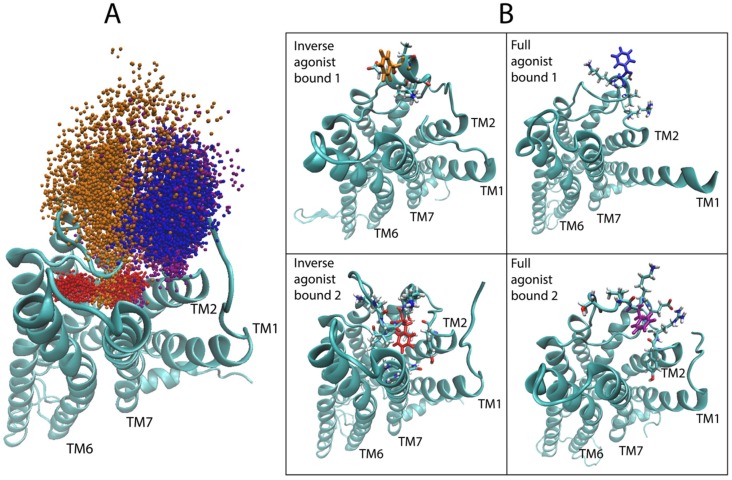
Conformational dynamics of residue F222 in ECL2 identified by the positions of the *epsilon* carbon atom Cε of F222 in the simulation trajectories. (**A**) Sampling of the coordinates of Cε of F222 (atom 2532), the top hit of the sensitivity analysis for serotonin (5-HT), in the aligned trajectories of the 5-HT_2A_R bound to the inverse agonist ketanserin (KET) (red and orange) and the full agonist 5-HT (blue and purple). (**B**) Representative frames from each of the four trajectories shown in the previous panel. F222 is colored solidly according to the bound ligand, ketanserin (red and orange) and serotonin (blue and purple).

**Figure 9 molecules-24-02097-f009:**
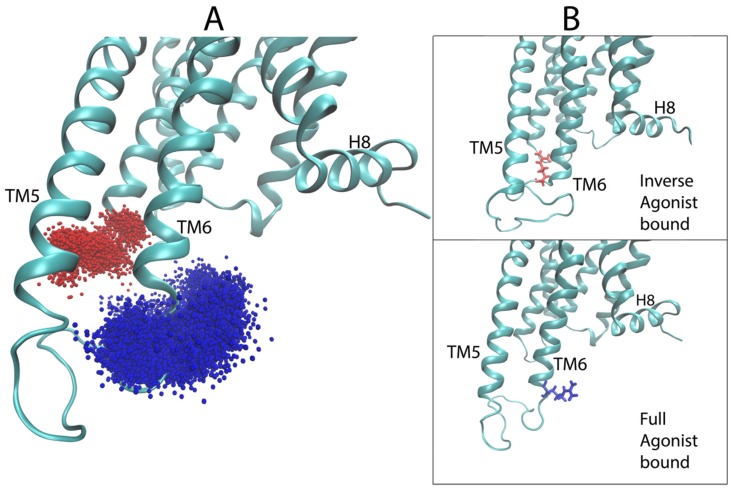
Conformational dynamics of residue R310 in ICL3 identified by the *epsilon* hydrogen highlighted by the sensitivity analysis. (**A**) Sampling of the coordinate of the epsilon hydrogen of R310 (atom 3535) in the ICL3, over all of the aligned trajectories of the 5-HT2AR bound to the inverse agonist ketanserin (red) and the full agonist serotonin (blue). (**B**) Example frames from each class of trajectories shown in panel A. R310 is colored solidly according to the bound ligand, KET (red) and 5-HT (blue).

**Figure 10 molecules-24-02097-f010:**
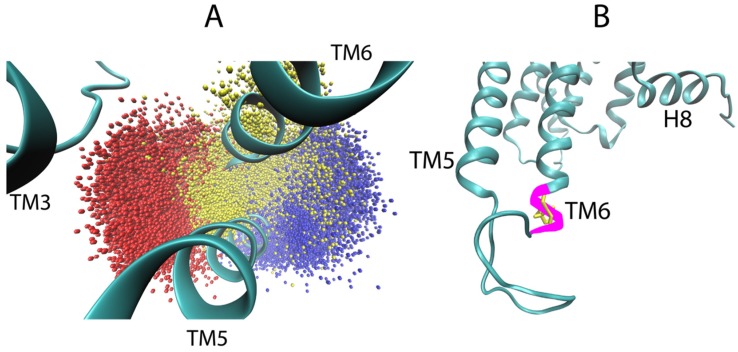
Conformation of the ICL3 of the 5-HT_2A_R in complexes with full- partial- and inverse-agonists. In most of the partial agonist trajectories, a piece of the ICL3 (magenta) forms an extra helical turn of TM6. In the full and inverse agonist-bound trajectories, this piece of the ICL3 is disordered/unfolded, with T307 pointing inwards towards TM3 in the inverse agonist trajectories and pointing away from TM3 in the full agonist-bound trajectories. (**A**) The sampling of the coordinate of the gamma oxygen of T307, an atom identified as important in the attention map of the partial agonist, for all the trajectories of the 5-HT_2A_R bound to full agonist (in blue), inverse agonist (red), and partial agonist (yellow). (**B**) Illustrative frame from partial agonist-bound trajectories showing the extended helical conformation of the ICL3. T307 is colored in yellow and the extra helical turn is colored in magenta.

**Figure 11 molecules-24-02097-f011:**
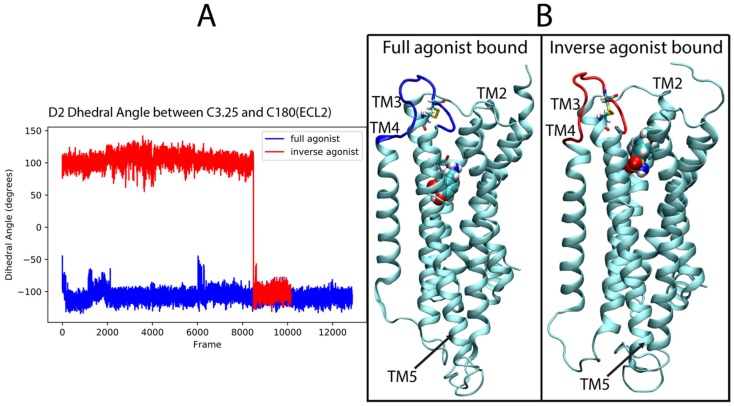
Involvement of residue C182 identified by the D2R sensitivity analysis in the discrimination of ECL2 structures. (**A**) The dihedral angle is measured along the disulfide bond between C182 and C3.25 for the full agonist-bound (blue) and inverse agonist-bound (red) trajectories. (**B**) Illustrative frames from the trajectories in panel A representing the two dihedral angle states. The ligand for each class (Dopamine for full agonist, and Sulpiride for the inverse agonist) is shown in function colored CPK.

**Figure 12 molecules-24-02097-f012:**
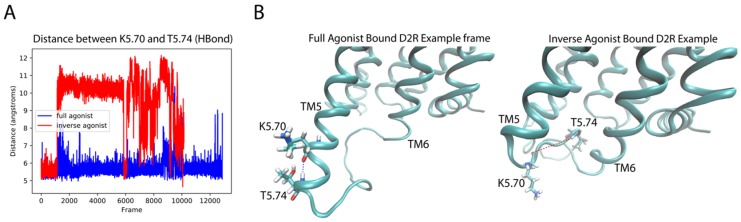
Residues K5.70 and T7.54 identified by the D2R sensitivity analysis determine the helicity of the intracellular TM5 extension. (**A**) The distance between the backbone oxygen of K5.70 and the backbone amide hydrogen of T5.74 along trajectories of D2R bound to the full agonist (in blue) and the inverse agonist (red). (**B**) Illustrative frames from the trajectories in panel A chosen for the distinct difference in the distance, and its effect on the helicity of the intracellular end of TM5.

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
