# Peer review of "A Machine Learning Approach for the Discovery of Ligand-Specific Functional Mechanisms of GPCRs"

_molecules, 2019, doi:10.3390/molecules24112097_

Round 1

Reviewer 1 Report

The authors reported an innovative approach to analyze GPCRs functional mechanisms.

The quality of the paper is good and in my opinion it is acceptable for publication after a discussion of the following points:

·       A broader description of the scrambling procedure should be added because it is not very clear.

·       Instead of applying the scrambling method, since GPCRs show a similar TM skeleton, could it be possible to align all the trajectories to a 7TM GPCR template and then convert the so-aligned atoms to pixels?

·       As with most computational studies, the results depend on the input data. In this study the results depends on the quality of docking/MD studies. A) The following phrase should be better discussed in the text: “A starting binding pose was chosen for each ligand based on the e-modal score and comparison to experimental data”. B) Furthermore, in the supporting information the trajectory analysis (alpha carbon and ligand heavy-atoms) of the ligand-protein complexes should be reported to better confirm the reliability of the techniques used.

·       The authors used only a total of 8 ligands, and this is one of the main limits of this study. The authors should include a larger number of ligands in order to better support the reliability of the obtained results.

Author Response

Thank you for the comprehensive review. Please see attached responses

Reviewer 2 Report

A good study demonstrating a novel approach for discovery of ligand-specific mechanisms of GPCRs. I have no comments.

Author Response

REVIEWER 2

A good study demonstrating a novel approach for discovery of ligand-specific mechanisms of GPCRs. I have no comments.

We are grateful to the Reviewer for welcoming our work.

Reviewer 3 Report

GPCRs are arguably the single most important drug target, and progress in drug design related to GPCRs is of very broad interest.  This paper shows how to use new object recognition methods for rational drug design targeting GPCRs. This paper from the team led by Harel Weinstein is an outstanding paper, important and timely in the current period of rapid progress on machine learning.  It is well suited to the scope of Molecules.

I especially appreciated the clarity of the writing. The introduction, for example, explains the background very clearly, and the algorithm is presented in clear detail. I think the scrambling procedure is brilliant; I myself have wrestled with that aspect of the problem, but certainly with less success.

In my experience, research directed to extracting hidden correlations from MD trajectories has been pursued for a long time but never quite as intriguingly as here. I recommend rapid publication without any changes.

I prefer that my reviews not be published, although of course they are available to authors, editors, and other reviewers.

Author Response

REVIEWER 3

GPCRs are arguably the single most important drug target, and progress in drug design related to GPCRs is of very broad interest.  This paper shows how to use new object recognition methods for rational drug design targeting GPCRs. This paper from the team led by Harel Weinstein is an outstanding paper, important and timely in the current period of rapid progress on machine learning.  It is well suited to the scope of Molecules.

We appreciate the Reviewer’s warm reception of our work.

I especially appreciated the clarity of the writing. The introduction, for example, explains the background very clearly, and the algorithm is presented in clear detail. I think the scrambling procedure is brilliant; I myself have wrestled with that aspect of the problem, but certainly with less success.

We are very glad that the Reviewer found the paper to be clear and instructive.

Round 2

Reviewer 1 Report

The paper can be accepted for pubblication